# Cardiomyocyte Damage: Ferroptosis Relation to Ischemia-Reperfusion Injury and Future Treatment Options

**DOI:** 10.3390/ijms241612846

**Published:** 2023-08-16

**Authors:** Jolanta Laukaitiene, Greta Gujyte, Edmundas Kadusevicius

**Affiliations:** 1Faculty of Medicine, Medical Academy, Lithuanian University of Health Sciences, 9 A. Mickeviciaus Street, LT-44307 Kaunas, Lithuania; jolanta.laukaitiene@lsmuni.lt; 2Cardiology Clinic, University Hospital, Lithuanian University of Health Sciences, Eiveniu Str. 2, LT-50161 Kaunas, Lithuania; greta.gujyte@lsmuni.lt; 3Institute of Physiology and Pharmacology, Medical Academy, Lithuanian University of Health Sciences, 9 A. Mickeviciaus Street, LT-44307 Kaunas, Lithuania

**Keywords:** atherosclerosis, ferroptosis, iron, myocardial infarction, ischemia-reperfusion injury, reactive oxygen species

## Abstract

About half a century ago, Eugene Braunwald, a father of modern cardiology, shared a revolutionary belief that “time is muscle”, which predetermined never-ending effort to preserve the unaffected myocardium. In connection to that, researchers are constantly trying to better comprehend the ongoing changes of the ischemic myocardium. As the latest studies show, metabolic changes after acute myocardial infarction (AMI) are inconsistent and depend on many constituents, which leads to many limitations and lack of unification. Nevertheless, one of the promising novel mechanistic approaches related to iron metabolism now plays an invaluable role in the ischemic heart research field. The heart, because of its high levels of oxygen consumption, is one of the most susceptible organs to iron-induced damage. In the past few years, a relatively new form of programmed cell death, called ferroptosis, has been gaining much attention in the context of myocardial infarction. This review will try to summarize the main novel metabolic pathways and show the pivotal limitations of the affected myocardium metabolomics.

## 1. Introduction

Most common cardiomyocyte injury is due to prolonged ischemia in the setting of an AMI. Timely restoration of blood by percutaneous coronary intervention and less-often-performed thrombolytic therapy is a standardized approach to saving the myocardium and limiting the size of the infarct. However, myocardial ischemia and processes followed by reperfusion also can induce the death of cardiomyocytes. A number of different studies in animal models of AMI suggest that the phenomenon called ischemia-reperfusion injury (IRI) accounts for up to 50% of the final myocardial infarct size [1], thereby reducing reperfusion beneficial effects. Currently used ferroptosis biomarkers (ACSL4, GPX4, iron, malondialdehyde) in preclinical studies are non-specific and present in other forms of cell death and certain pathological conditions [2]. Recent studies regarding altered-iron homeostasis in cardiomyocytes have shown that it directly induces ferroptosis by accumulation of phospholipid hydroperoxides in the cell membrane. Presently, there is no available therapeutic approach to prevent restored-blood-flow-induced injury; therefore, there is need to explore the pathophysiology to improve clinical outcomes.

## 2. Iron Metabolism in the Heart

Iron metabolism in the heart plays a crucial role in maintaining cellular functions and homeostasis. Cardiomyocytes rely on a complex process of iron uptake and recycling to ensure an adequate supply of this essential micronutrient.

The cellular iron uptake in cardiomyocytes is dependent on the endocytosis of diferric transferrin (TF) bound to its receptor transferrin receptor protein 1 (TFR1) [3]. The endosome is then acidified by vacuolar ATPase, which leads to the reduction of ferric iron (Fe^3+^) to ferrous iron (Fe^2+^) by the six-transmembrane epithelial antigen of prostate (STEAP) family of metalloreductases [1,4]. The Fenton reaction converts Fe^2+^ into Fe^3+^, which induces lipid peroxidation by activating lipoxygenases. Oxidized Fe^2+^ is then released from the endosome into the cytoplasm via natural resistance-associated macrophage protein 2 (NRAMP2; also known as DMT1), and apo-TF and TFR1 are shuttled back to the cell surface to be reused by the cell [3,5]. Excess iron either binds to the heavy chain of ferritin (FTH) or is exported by ferroportin (FPN). In addition, iron can be released from FTH via nuclear receptor coactivator 4 (NCOA4)-mediated autophagic degradation of lysosomal ferritin, a process known as ferritinophagy [6].

This intricate system of iron metabolism in the heart ensures a delicate balance between iron uptake, storage, and release. Dysregulation of this process can lead to iron overload or deficiency, both of which can have detrimental effects on cardiac function and overall health. Further research into the regulation and dynamics of iron metabolism in the heart is crucial for understanding and potentially treating iron-related disorders and cardiac conditions.

## 3. Mitochondrial Iron Regulation

Mitochondria serve as the primary site for the utilization and accumulation of iron, which is essential for sustaining various physiological functions. Once iron is transported into the mitochondria, it is utilized in the synthesis of cofactors that play critical roles in enzyme functions associated with redox reactions, energy metabolism, DNA synthesis, and other essential cellular processes. Iron undergoes crucial metabolic processes primarily involving three pathways: heme synthesis, Fe-S cluster biogenesis, and mitochondrial iron storage (Figure 1).

### 3.1. Heme Synthesis

Heme, a crucial cofactor in various essential proteins such as catalases, peroxidases, and cytochrome P450, consists of ferrous iron and protoporphyrin IX (PPIX). It plays vital roles in oxygen storage and transport, signal transduction, enzyme redox reactions involving electron transfer, and gene expression regulation. Heme proteins can also serve as sensors for diatomic gases. In mammals, the synthesis of heme heavily relies on mitochondrial activity and involves eight sequential enzymes, four in the cytosol and four in the mitochondria. This biosynthetic pathway spans both the cytoplasm and mitochondrial compartments. The initial enzyme in heme biosynthesis is 5-aminolevulinic acid synthase (ALAS), located in the mitochondrial matrix, which catalyzes the condensation of glycine and succinyl coenzyme A (CoA) to produce 5-aminolevulinic acid (ALA) [7]. Notably, there are two isoforms of ALAS: ALAS1, expressed widely in all cells, and ALAS2, found exclusively in red blood cells. ALAS2 expression is regulated by the IRP-IRE system due to the presence of an IRE in its 5′ UTR. This distinction suggests that heme synthesis in ALAS2-expressing erythroid cells may be influenced by the availability of cytoplasmic iron. The final enzyme in the heme biosynthesis pathway is ferrochelatase, responsible for incorporating ferrous iron into PPIX to form heme. Interestingly, mammalian ferrochelatase contains Fe-S clusters, indicating that impaired Fe-S cluster synthesis might affect the rate of heme synthesis as well [8].

### 3.2. Fe-S Cluster Biogenesis

Mitochondria serve as the main generators of both heme and iron–sulfur (Fe-S) clusters, which are essential iron-containing cofactors. Fe-S clusters, composed of inorganic iron and sulfur, are ancient and functional prosthetic groups found in numerous proteins. The formation of core Fe-S clusters in humans involves cysteine desulfurase (NFS1), which extracts sulfur from cysteine and is critical for alanine production, and scaffold protein (IscU), an iron–sulfur cluster assembly protein that provides protein scaffolding and plays a vital role in electron transport and oxidation–reduction reactions. Fe-S clusters exhibit various functional structures, such as rhomboid [2Fe–2S], cuboidal [3Fe–4S], and cubane [4Fe–4S] configurations, determined by different iron and sulfur atom quantities. These clusters are typically linked to cysteine residues, necessary for protein synthesis, folding, enzyme catalysis, interconversion of small molecules, and oxygen sensing. In some cases, other amino acids can also act as ligands. Fe-S clusters are vital for functional electron transfer in the respiratory chain complex and play an important role in DNA helicase, metabolism, and DNA transcription [8].

A significant amount of cellular iron is transported into the mitochondria because they are the primary site of heme and iron–sulfur (Fe-S) cluster biosynthesis [9]. Iron is transported across the mitochondrial membrane by mitoferrins 1 and 2, which are also known as SLC25A37 and SLC25A28, respectively [6]. Although the exact mechanism of mitochondrial iron export is unknown, iron is probably exported from mitochondria either in the form of Fe-S clusters, heme, or iron conjugated to glutathione (GSH), but not as free iron. To stop the production of reactive oxygen species (ROS), mitochondrial iron is retained by interacting with mitochondria-specific ferritin (FTMT). Mutations in FTMT result in mitochondrial iron overload and cytoplasmic iron deficiency, demonstrating that FTMT functions actively in iron transport to the cytoplasm in addition to mitochondrial iron storage [10,11]. While the export of Fe-S clusters into the cytoplasm may require the Fe-S cluster transporters ABCB7 (mitochondrial) and ABCB8 (also known as the mitochondrial potassium channel ATP-binding subunit), FLVCR1B (feline leukemia virus subgroup C receptor-related protein 1B) promotes heme efflux into the cytoplasm. Furthermore, it has been established that the ATP-binding cassette subfamily B member 10, mitochondrial (ABCB10) controls the initial stages of heme synthesis in mitochondria [2,11,12].

### 3.3. Mitochondrial Iron Storage

Mitochondria serve as hubs of high redox activity and act as the primary source of reactive oxygen species (ROS) generation. To prevent ROS production and the detrimental Fenton reaction, mitochondria rely on a continuous uptake of iron for the synthesis of heme and Fe-S clusters. Mitochondrial iron can follow two paths: it is either utilized in various synthesis pathways or stored within ferritin (FTMT). FTMT, encoded by an intronless gene located on human chromosome 5q23.1, has the unique ability to store iron in homopolymers and exhibits distinct iron oxidation and hydrolysis chemistry compared with H-ferritin. The expression of FTMT is specific to certain cell types and organs, with the highest levels found in human testes, while being absent in the liver and spleen, despite its critical role in iron storage. A crucial function of FTMT is to prevent iron-induced ROS production in mitochondria by sequestering free iron. However, excessive expression of FTMT can disrupt the delicate balance of cellular iron levels, indicating its significant role in regulating systemic iron homeostasis [13,14].

## 4. Cardiac Ferroptosis

Iron metabolism, lipid peroxidation, and antioxidant systems are the main components involved in the mechanism of ferroptosis. In eukaryotic cells, mitochondria serve as the primary source of energy and coordinate crucial metabolic processes like oxidative phosphorylation. Given that the mitochondria are the primary site for the synthesis of heme and Fe-S clusters, the mitochondria are rich in iron [15].

The central role of lipid peroxidation in driving ferroptotic cell death indicates that ferroptosis can be caused by the collapse of the GSH-glutathione peroxidase 4 (GPX4) antioxidant systems. System xc– is a heterodimeric transmembrane complex composed of a light chain, solute carrier family 7 member 11 (SLC7A11/xCT), and a heavy chain, solute carrier family 3 member 2 (SLC3A2) [2,5,16]. After entering the cells by system xc–, cystine is quickly reduced to cysteine, which is mainly utilized for the synthesis of GSH [17,18]. A depletion of cysteine levels results in intracellular GSH downregulation and weakens the activity of GPX4, which is responsible for the conversion of toxic polyunsaturated lipid hydroperoxides (L –OOHs) to non-toxic and less reactive lipid alcohols (L-OHs). Consequently, loss of GPX4 expression or GPX4 inactivation leads to L-OOH accumulation, which interacts with excessive Fe^2+^ and contributes to lipid peroxides that are harmful to nucleic acid and proteins [19,20]. The pharmacological inhibitors of system xc– (e.g., erastin) and GPX4 (e.g., RSL3) are the classical two ferroptosis inducers. In addition, several GPX4-independent anti-ferroptosis pathways have recently been identified, such as the apoptosis-inducing factor mitochondria-associated 2 (AIFM2)-mediated CoQ10 production pathway, and the endosomal sorting complex required for the transport-III (ESCRT-III)-dependent membrane repair pathway [21,22]. These findings indicate that multiple antioxidants and membrane repair pathways limit the oxidative damage caused by ferroptosis, although their selectivity and specificity in ferroptosis are still unclear. ROS interact with polyunsaturated fatty acids (PUFAs) of lipid membranes and induce lipid peroxidation. Acyl-CoA synthetase long-chain family member 4 (ACSL4) is involved in the biosynthesis of PUFAs and helps free PUFAs to be esterified and incorporated into membrane phospholipids [23,24]. The release of 4-hydroxynonenal (4-HNE) and malondialdehyde (MDA) following membrane lipid peroxidation causes cellular structural damage.

A recent study demonstrated that ferroptosis occurs mainly in the phase of myocardial reperfusion but not ischemia. The ferroptosis was evaluated with the levels of ACSL4, GPX4, iron, and MDA [17,25,26]. An iron chelator (deferoxamine) was used to verify the contribution of ferroptosis on IRI. The results showed that ischemic damage (infarction and CK release) worsened with the extension of ischemia time, but there were no significant changes in the ferroptosis indexes (ACSL4, GPX4, iron, and MDA) in cardiac tissues [27]. In contrast, the levels of ACSL4, iron, MDA were gradually elevated with the extension of reperfusion concomitant with a decrease in GPX4 levels (Figure 2). All mechanisms mentioned above were specific to cardiomyocytes, although most mechanisms were shared with other cells as well.

## 5. Ferroptosis in Medication-Induced Myocardial Injury

Anthracyclines are a group of chemotherapeutic drugs that are used to treat many types of malignancies. While these antineoplastic drugs are extremely effective in treating cancer, they can also cause significant cardiotoxicity, leading to congestive heart failure. The risk of cardiotoxicity increases with the dose and duration of treatment, and patients who receive anthracyclines are typically monitored for signs of cardiotoxicity. For example, to prevent cardiomyopathy, a cumulative dose of 450–550 mg/m^2^ body surface area of doxorubicin (DOX) is recommended [21,25,28]. Although the exact mechanism causing doxorubicin-induced cardiomyopathy is unknown, new evidence points to iron–related damage. One of the mechanisms by which anthracyclines cause cardiotoxicity is through the development of IRI [4,29,30].

The most-studied drug of the anthracycline class in terms of IRI is DOX. It has been suggested that DOX induces an iron-mediated increase in ROS. Regarding how DOX affects iron metabolism in the heart, one study discovered that iron accumulates specifically in the mitochondria of doxorubicin-treated cardiomyocytes as a result of suppression of the mitochondrial iron exporter ABCB8 [25,31,32]. Cardiotoxicity caused by DOX is prevented by overexpressing ABCB8 (ABCB8 overexpression reduces DOX-induced mitochondrial iron accumulation and cardiotoxicity) or directly chelating mitochondrial iron with the iron chelator dexrazoxane [29,33]. Furthermore, DOX treatment in mice has been shown to increase the expression of cardiac mitochondrial ferritin, and genetic deletion of mitochondrial ferritin increased the susceptibility of cardiomyocytes to iron toxicity brought on by DOX. Also, a subsequent study that was released in 2020 [21] showed that DOX treatment can downregulate GPX4 expression in the heart, leading to excessive lipid peroxidation. However, the exact mechanism by which doxorubicin treatment can suppress GPX4 needs further investigation [34].

Research on the cardiotoxicity of anticancer agents is still in its early stages due to the relatively new mechanism of cell death. Recent studies mainly present the fact that antineoplastic medications cause cardiotoxicity through ferroptosis, but there is a lack of additional mechanistic material. In the research field, other than doxorubicin, medications such as fluorouracil, imatinib, trastuzumab are being tested, but there is very little knowledge of mechanistic approaches.

## 6. Ferroptosis Role in Heart Failure

Due to the pervasiveness of lipid oxidation, ferroptosis is a constantly expanding field that is connected to numerous other areas of study. New models of heart failure with preserved ejection fraction (HFpEF) are beginning to clarify a connection between HFpEF and ferroptosis.

In HFpEF, vascular endothelium damage is a crucial initial event that sets off myocardial inflammation. Additionally, it has been determined that iron overload contributes to endothelial dysfunction by encouraging an excessive amount of ROS production. An insufficient antioxidant response and increased ROS production result in oxidative damage, which promotes the initiation of HFpEF [35]. The formation of superoxide and hydroxyl radicals, the latter of which is the most potent and toxic, is triggered by iron, which plays a key role in this process. It is important to remember that redox-active iron is required for the activity of enzymes, which play a role in the production of ROS, such as NOX and NOS, and are important in the development of heart failure with preserved ejection fraction [36,37].

Furthermore, by recycling iron via phagocytosis and releasing it back into circulation, macrophages play a crucial part in preserving iron homeostasis in myocardial tissue. Heme protein release in heart failure cases causes an increase in iron sources, which results in iron overload in macrophages. By activating M1 macrophages, which tends to cause inflammation, while depleting anti-inflammatory M2 macrophages, this iron overload causes myocardial inflammation and accelerates the development of heart failure. Although some studies have shown conflicting effects of iron deficiency or excess on macrophages during inflammation, new evidence suggests that iron overload may actually be responsible for the development of myocardial fibrosis in patients with heart failure, suggesting that iron chelation may be an effective treatment option [38,39].

Additionally, iron has an impact on cardiomyocytes and calcium signaling, which are crucial for proper diastolic relaxation and systolic cardiac contraction. As a result of serum iron levels’ detrimental effects on calcium transporters, the expression of the transferrin receptor is lowered to protect against iron overload. Additionally, the transport of myocardial non-transferrin-bound iron (NTBI) is significantly influenced by serum iron levels, which control the L-type calcium channels (LTCCs). However, the excitation-contraction coupling (ECC) of HFpEF cardiomyocytes is hampered by the iron-induced slowing of LTCC current inactivation, which increases ionized calcium entry and overload [40]. 

## 7. Treatment Options

Oxidative stress plays an important role in the pathological process of the ferroptotic mechanism in myocardial injury. Iron is a crucial element required for a variety of vital biological functions, but dysregulated iron homeostasis also has potentially toxic effects.

Targeting ferroptosis mechanisms may be a promising therapeutic option to prevent myocardial ischemia-reperfusion injury and to improve the survival and prognosis of myocardial infarction patients. In the case of IRI, excessive iron is transported into cells that make cardiomyocytes more vulnerable by the accumulation of ROS through the Fenton reaction and Haber–Weiss reaction. Thus, it is supposed that ferroptosis inhibitors are expected to prevent MIRI deterioration by suppressing cardiomyocyte ferroptosis and can be considered a potential target for cardiovascular diseases pharmacotherapy [41,42,43,44,45]. 

Novel chemical and/or biological entities related to pathophysiological alterations from iron homeostasis to ferroptosis, together with potential pathways regarding ferroptosis secondary to cardiovascular IRI, could be potential targets for the development of new drugs for the management of myocardial ischemia-reperfusion injury. Most of chemical and/or biological entities that are able to suppress ferroptosis induced by RSL3 or erastin or so-called ”ferroptosis inhibitors“ can be classified as antioxidants (Ferrostatin-1, Liproxstatin-1, Zileuton, Nuclear factor erythroid 2-related factor 2, vitamins C and E, Xanthohumol (XH), Ferulic acid (FA), Resveratrol (Res), Cyanidin-3-glucoside (C3G)), iron chelators (Histochrome, Deferoxamine, Dexrazoxane), and ROS scavengers (N-acetyl-L-cysteine, XJB-5-131, JP4-039, and mitoquinone) (Table 1). 

Most of the potential ferroptosis inhibitors were tested in in vivo and in vitro studies, and only a few of them were used in randomized clinical trials (RCT). The most popular method to study ischemia/reperfusion (I/R) is the Langendorff model (ex vivo), which consists of a functionally active isolated heart having a cannula inserted into its aorta so that the heart can be retrogradely perfused via the coronary circulation. In vivo studies have been performed in the hearts of living organisms of species, such as, in most cases, mouse and rat. Cardiac tissue cell cultures have also been subjected to study ferroptosis inhibitors in I/R settings.

**Table 1 ijms-24-12846-t001:** Ferroptosis inhibitors.

Compound	Source	Testing Model	Mechanism of Action	References
N-Acetylcysteine (NAC)	Synthetic antioxidant	RCT	Prodrug to L-cysteine, a precursor to the biologic antioxidant glutathione, an ROS scavenger, indirect action as a metal ion chelator and ability to inhibit NF-_B	[46,47,48,49]
Vitamin C (vit C)	Water-soluble vitamin found in citrus and other fruits and vegetables	RCT	Has action as an ROS scavenger	[50,51]
Vitamin E (Vit E)	Fat soluble vitamin a group of eight fat-soluble compounds including four tocopherols and four tocotrienols	RCT	Has action as an ROS scavenger	[52,53,54]
Deferoxamine (DFO)	Synthetic iron-chelating agent	RCT	Acts by binding free iron in the bloodstream and enhancing its elimination in the urine	[55,56]
Liproxstatin-1 (Lip-1)	Synthetic antioxidant	Langendorff model using male C57BL/6J mice hearts (ex vivo)	Radical-trapping antioxidant (RTA) via reduction in voltage-dependent anion channel 1 (VDAC1)	[57,58]
Ferrostatin-1 (Fer-1)	Synthetic antioxidant	Cell models (ex vivo)	Radical-trapping antioxidant (RTAs) via GPX4 inhibition, GPX4 deletion, or GSH depletion and ability to inhibit lipid peroxidation directly by trapping chain-carrying radicals	[57,59]
Zileuton (ZIL)	Synthetic derivative of hydroxyurea	HepG2 and HL60 cells models (ex vivo)	Leukotriene inhibitor, blocks 5-lipoxygenase and formation of 5-HETE in ACSL4-overexpressed cells	[5,60]
Nuclear factor erythroid 2-related factor 2 (NRF2)	Transcription factor in humans encoded by the NFE2L2 gene	HUVECs and EA.hy926 vascular endothelial cell models (ex vivo)	Regulates the expression of antioxidant proteins that protect against oxidative damage	[25,61]
Xanthohumol (XH)	A natural product found in the female inflorescences of Humulus lupulus, also known as hops	Neonatal rat cardiomyocytes (NRCMs) model (ex vivo)	Has action as an ROS scavenger, chelate Fe^2+^, and regulates NRF2 and GPX4 protein levels in cardiomyocytes during Fe-SP and RSL3-induced ferroptosis	[29]
Ferulic acid (FA)	A natural organic polyphenol derived from the genus Ferula	Murine MIN6 cells models (ex vivo)	Enhances the activity of antioxidant enzymes (SOD, GSH-Px, and CAT)	[62,63]
Resveratrol (Res)	A stilbenoid, a type of natural phenol, and a phytoalexin produced by several plants in response to injury	Male rats (Sprague-Dawley) (in vivo)	Blocks oxidative stress and Fe^2+^ levels in IR models and regulates USP19-Beclin 1 autophagy	[21,64]
Gossypol acetic acid (GAA)	A natural product isolated from cottonseeds and roots	mitochondria from the *Mcu*^−/−^, *Mcu^fl/fl–MCM^*, and DN-*Mcu* mouse models (ex vivo)	Downregulates PTGS2 and ACSL4 levels in both mRNA and protein	[65,66]
Naringenin (NAR)	A natural flavanone compound, widely distributed in several citrus fruits	AKI mouse model (ex vivo)	Downregulates NRF2	[61]
Cyanidin-3-glucoside (C3G)	A natural anthocyanin polyphenol with polyphenolic structure widely occurring in plants	H9c2 cells (ex vivo)	Can relieve oxidative stress, downregulate LC3II/LC3I and TFR1 levels, and upregulate FTH1 and GPX4 expression	[67,68]
Histochrome (HC)	Isolated from sea urchin Scaphechinus and standardized echinochrome	AKI mouse model (ex vivo)	Reduces cytosolic and mitochondrial ROS, maintains intracellular GSH levels, and elevates GPX4 activity	[17,69]
Dexmedetomidine (Dex)	A synthetic sympatholytic medicine agonist of α_2_-adrenergic receptors in the brain	EA.hy926 vascular endothelial cells (ex vivo)	Chelates iron and activates NRF2 through the AMPK/GSK-3β pathway	[28,61,65]
Dexrazoxane (DXZ),	A derivative of Ethylenediaminetetraacetic acid (EDTA)	(Mlkl^−/−^ and Fadd^−/−^Mlkl^−/−^) mice model (in vivo)	Chelates iron and regulates the level of Ptgs2 mRNA	[70]
XJB-5-131	A synthetic antioxidant	HT-1080, BJeLR, and panc-1 cells (ex vivo)	Nitroxide-based lipid peroxidation, ROS scavengers	[71]
JP4-039	A synthetic antioxidant	HT-1080, BJeLR, and panc-1 cells (ex vivo)	Nitroxide-based lipid peroxidation, ROS scavengers	[71]

## 8. Antioxidants

### 8.1. Liproxstatin-1 (Lip-1)

Efforts to prevent ferroptosis in pathological settings have therefore focused on the use of radical trapping antioxidants (RTAs), such as liproxstatin-1 (Lip-1) acting as a lipid peroxide scavenger [3,46]. Lip-1 administrated after an ischemic event provides cardioprotection by reducing the extent of myocardial damage and maintaining the integrity of the mitochondrial structure. Lip-1 cardiac protection involves a reduction in voltage-dependent anion channel 1 (VDAC1) levels and oligomerization, without affecting VDAC2/3. For a long time, VDAC1 was considered to take a central role in inducing cell death by opening the mitochondrial permeability transition pore (mPTP), but latterly, Yansheng et al. showed that GPX4 does not impact Ca^2+^-induced mPTP opening [20,47]. Moreover, Lip-1 administration restored cytosolic antioxidant GPX4 depletion and limited ROS production in mitochondria by catalyzing the reaction, which reduces phospholipid hydroperoxides to less harmful alcohols [58]. 

### 8.2. Ferrostatin-1 (Fer-1)

Ferrostatin-1 (Fer-1), a synthetic antioxidant, a potent and selective ferroptosis inhibitor, suppresses Erastin-induced ferroptosis in HT-1080 cells (EC_50_ = 60 nM). Fer-1 acts via a reductive mechanism to prevent damage to membrane lipids and thereby inhibits cell death. The anti-ferroptotic activity of Fer-1 is due to the scavenging of initiating alkoxyl radicals produced, together with other rearrangement products, by ferrous iron from lipid hydroperoxides, and the Fer-1 inhibits ferroptosis much more efficiently than phenolic antioxidants [57]. This activity likely derives from their reactivity as RTAs rather than their potency as inhibitors of lipoxygenases. Although inhibited autoxidations of styrene revealed that Fer-1 and Lip-1 react roughly 10-fold more slowly with peroxyl radicals than reactions of α-tocopherol (α-TOH), they were significantly more reactive than α-TOH in phosphatidylcholine lipid bilayers—consistent with the greater potency of Fer-1 and Lip-1 relative to α-TOH as inhibitors of ferroptosis [47,49,50]. Targeted ferroptosis can potentially provide preventative treatment of MIRI in patients undergoing heart transplantation after coronary artery reperfusion, since Fer-1 reduces myocardial cell death and blocks the recruitment of neutrophil granulocytes to damaged myocardial cells by damage-associated molecular patterns (DAMPs) after heart transplantation [50].

### 8.3. Zileuton (ZIL)

The leukotriene inhibitor Zileuton is a synthetic derivative of hydroxyurea. ZIL blocks 5-lipoxygenase, which catalyzes the formation of leukotrienes from arachidonic acid and produces bronchodilation with decreases in bronchial mucous secretion and edema. ZIL is used to prevent or to decrease the symptoms of asthma. Despite its use for the management of asthma, ZIL, a pharmacological inhibitor of 5-HETE, could block ferroptosis in ACSL4-overexpressed cells and effect regulation of lipid metabolism pathways in cardiovascular diseases [42,52].

### 8.4. Nuclear Factor Erythroid 2-Related Factor 2 (NRF2) 

Nuclear factor erythroid 2-related factor 2 (NRF2), also known as nuclear factor erythroid-derived 2-like, is a transcription factor that in humans is encoded by the NFE2L2 gene. NRF2 is a basic leucine zipper (bZIP) protein that may regulate the expression of antioxidant proteins that protect against oxidative damage triggered by injury and inflammation, according to preliminary research. Similarly, britanin upregulates GPX4 expression via same pathway, and the knockdown of NRF2 blocks the protective effects of britanin against IRI-induced damage in H9C2 cells [28,53]. 

### 8.5. Vitamin E

Vitamin E, which is fat soluble and one of the most potent antioxidants, is an important agent in the prevention of cardiovascular diseases. Studies have confirmed that a higher α-TOH baseline serum concentration is associated with a decreased risk in overall and cause-specific mortality due to cardiovascular and heart diseases [54]. Vitamin E action to reduce damage from cardiac reperfusion ischemia has been demonstrated in animal cardioprotection models [55,56].

### 8.6. Vitamin C

Vitamin C is a water-soluble antioxidant medication and has action as an ROS scavenger. It has been demonstrated that vitamin C’s cardioprotective effect against cardiac IRI depends on its route of administration and corresponds to the concentration of vitamin C given to patients. The higher serum concentration and the most potential cardioprotective effect against oxidative stress induced by IRI was demonstrated with the intravenous formulation in comparison with oral conventional and oral liposome-encapsulated formulation [57,58]. Data from recent clinical trials showed that the intravenous infusion of vitamin C prior to PCI reduced cardiac injury biomarkers, as well as inflammatory biomarkers and ROS production. Improvements in functional parameters, such as left ventricular ejection fraction (LVEF) and telediastolic left ventricular volume, showed a trend but had an inconclusive association with vitamin C and its pharmacokinetic data. Therefore, it seems reasonable that these beneficial effects could be further enhanced by the association with other antioxidant agents [54].

## 9. Natural Herbal Phytochemicals

Some natural phytochemical compounds, derived from various herbals demonstrated ferroptosis inhibition properties on preclinical models and clinical trials. 

### 9.1. Xanthohumol (XH)

Xanthohumol (XH) is a natural product found in the female inflorescences of Humulus lupulus, also known as hops. This compound is also found in beer and belongs to a class of compounds that contribute to the bitterness and flavor of hops. The antioxidant effect of xanthohumol can reduce the generation of lipid peroxide and ROS, chelate Fe^2+^, and regulate NRF2 and GPX4 protein levels in cardiomyocytes during Fe-SP and RSL3-induced ferroptosis [29]. 

### 9.2. Ferulic Acid (FA)

Ferulic acid (FA), an organic compound phytochemical polyphenol derived from the genus Ferula, referring to the giant fennel (Ferula communis), is found in plant cell walls, covalently bonded to hemicelluloses such as arabinoxylans. In a study by Liu et al., FA facilitated energy production and decreased the AMP/ATP ratio by upregulating AMPKα2 expression, as well as inhibiting ferroptosis by enhancing the activity of antioxidant enzymes (SOD, GSH-Px, and CAT), which was similar to the effect of ferroptosis inhibitor Fer-1 [62]. Resveratrol (Res), a polyphenol with multiple bioactivities, was demonstrated to diminish oxidative stress and Fe^2+^ levels in IR models and regulate USP19-Beclin 1 autophagy to inhibit ferroptosis [21]. 

### 9.3. Cyanidin-3-Glucoside (C3G)

Cyanidin-3-glucoside (C3G) is a well-known natural anthocyanin and possesses antioxidant and anti-inflammatory properties. C3G treatment can relieve oxidative stress, downregulate LC3II/LC3I and TFR1 levels, and upregulate FTH1 and GPX4 expression in oxygen–glucose deprivation/reoxygenation (OGD/R)-treated H9c2 cells. In addition to inhibiting USP19 and LC3II protein levels, C3G enhances the K11-linked ubiquitination of Beclin 1 [10,53]. Cyanidin-3-glucoside (C3G) treatment can effectively alleviate the expression of proteins related to apoptosis, reduce Fe^2+^ content, and improve MIRI. Therefore, C3G is a potential medicine to prevent myocardial cells from being affected by MIRI [61,62].

### 9.4. Gossypol Acetic Acid (GAA) 

Gossypol acetic acid (GAA) a natural product isolated from cottonseeds and roots. Gossypol, as a PAF antagonist/inhibitor, markedly inhibited the contractile responses of guinea pig lung parenchyma strips stimulated with leukotriene B4, leukotriene D4, and PAF-acether and exerts cytoprotective effects by inhibiting ferroptosis via downregulation of PTGS2 and ACSL4 levels in both mRNA and protein. GAA remarkably reduces myocardial infarct size, decreases lipid peroxidation, activates NRF2, and downregulates levels of PTGS2 and ACSL4 in both mRNA and protein [53,63,64]. In addition, etomidate activates NRF2/HO-1 by promoting nuclear translocation of NRF2 to suppress IRI-induced ferroptosis and attenuate heart failure, pathological injury, myocardial fibrosis, and inflammation [53,64]. 

### 9.5. Naringenin (NAR) 

Naringenin (NAR), a natural flavanone compound widely distributed in several citrus fruits, bergamot, tomatoes, and other fruits, has received increasing attention in recent years due to its diverse pharmacological activities. However, this bioflavonoid shows poor solubility and therefore is difficult to absorb on oral ingestion. NAR attenuates histopathological injury, inflammation, and lipid peroxidation in heart tissue treated with IRI by regulating NRF2. Erastin inhibits the ability of NAR to protect H9C2 cardiomyocytes exposed to IRI [16,65].

However, despite the huge interest in antioxidant vitamins and natural herbal phytochemicals as potential protective agents against the development of myocardial IRI, the actual contributions and mechanisms of such compounds remain unclear, and results are debatable.

## 10. Iron Chelators

Histochrome (HC) has potent antioxidant content and the ability to form iron chelate [68]. The hearts of rats administered early intravenous injections of HC before reperfusion showed a marked reduction in cardiac fibrosis and an increase in capillary density. By inducing the expression of NRF2 and antioxidant genes, HC can reduce cytosolic and mitochondrial ROS, maintain intracellular GSH levels, and elevate GPX4 activity [17,67,68]. Some studies demonstrated that dexmedetomidine (Dex) significantly reduced myocardial infarction and improved heart function, along with diminished lipid peroxidation and Fe^2+^ accumulation. Also, studies have confirmed that Dex activates NRF2 through the AMPK/GSK-3β pathway to protect the heart from IRI-induced ferroptosis. 

Dexrazoxane (DXZ), a derivative of EDTA, chelates iron and thus reduces the number of metal ions complexed with anthracycline and, consequently, decreases the formation of superoxide radicals. The exact chelation mechanism is unknown, but dexrazoxane significantly increased the expression level of Ptgs2 mRNA, which further led to a reduction in the myocardial enzyme spectrum and the scar area of myocardial infarction [70]. A recent study found that dexrazoxane or ponatinib inhibited ferroptosis during MIRI, and a combined treatment with both drugs markedly reduced the scar area of myocardial infarction. Based on these findings, a combined treatment targeting different types of cell death is proposed as an effective treatment strategy for MIRI.

## 11. ROS Scavengers

Novel synthetic compounds defined as ROS scavengers, including XJB-5-131 and JP4-039, are reported to suppress ferroptosis via eliminating poisonous ROS. Their biological activity correlated well over several orders of magnitude with their structure, relative lipophilicity, and respective enrichment in mitochondria, revealing a critical role of intramitochondrial lipid peroxidation in ferroptosis. These results also suggest that preventing mitochondrial lipid oxidation might offer a viable therapeutic opportunity in ischemia/reperfusion-induced tissue injury [71].

## 12. Conclusions

More than a two decades ago, a theory linking iron to cardiovascular disease was initially proposed, but the mechanistic pathways underlying this relationship remained completely unknown until the discovery of ferroptosis, an iron-dependent form of regulated cell death, just 10 years ago. In recent years, studies on the role of ferroptosis in various diseases, and especially cardiovascular diseases, receive extensive attention. Now it is indisputable that ferroptosis is induced by the activation of iron-dependent lipid peroxidation, but the key effector molecules involved in this process are still unclear. For the treatment of patients with iron-overload-related cardiomyopathy, iron chelation therapy has been strongly advised due to the potential tissue damage that high levels of free reactive iron can cause. More targets have been identified for the activation and inhibition of ferroptosis in IRI, but it is still unclear which mechanism is dominant and has the best therapeutic impact. The creation of efficient ferroptosis-specific antagonists for clinical testing is being facilitated by preclinical research.

## Figures and Tables

**Figure 1 ijms-24-12846-f001:**
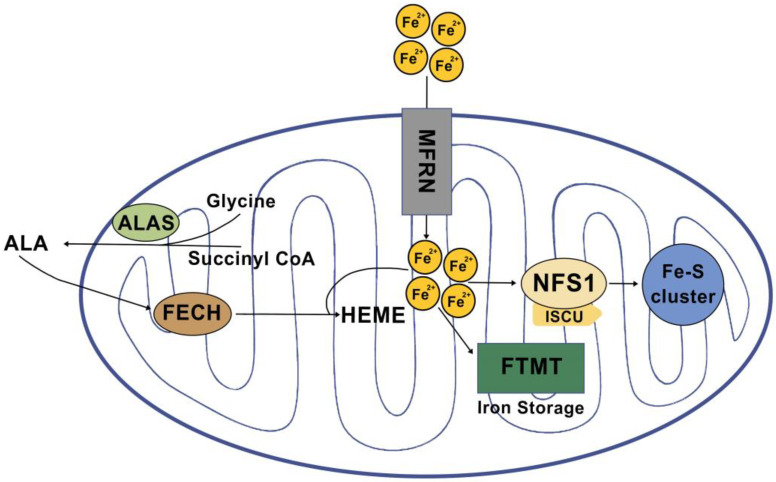
Mitochondrial iron metabolic pathways (based on in vitro model).

**Figure 2 ijms-24-12846-f002:**
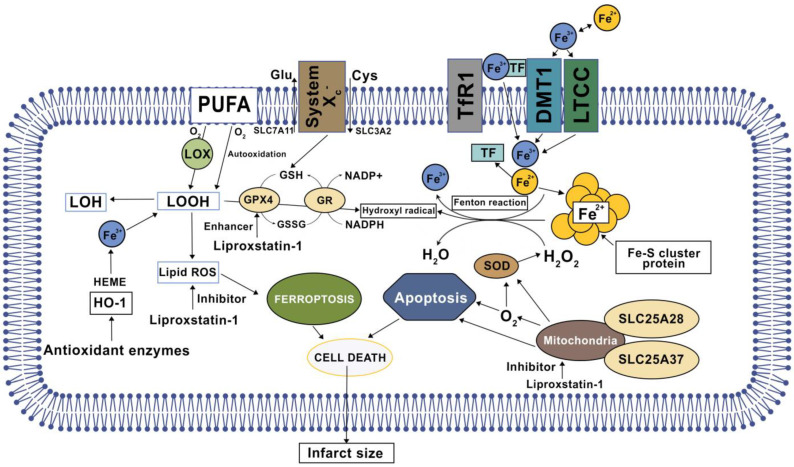
An overview of the mechanism and key regulators of ferroptosis (based on in vivo and in vitro models of IRI).

## Data Availability

Not applicable.

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
