# Peer review of "Cardiomyocyte Damage: Ferroptosis Relation to Ischemia-Reperfusion Injury and Future Treatment Options"

_ijms, 2023, doi:10.3390/ijms241612846_

Round 1
Reviewer 1 Report
I think this is a very extensive review covering the mechanism of ischemia-reperfusion injury as explained through the lenses of ferroptosis. This is a niche within the spectrum of cardiac failure that does not receive enough clinical attention. In that regard, this review is valuable. I only have some minor comments:
1. Authors should dedicate more space to the role of iron in heart failure.
2. Are there any treatments based on a ferroptosis-blocking mechanism that could be translated to acute coronary syndrome?
Nothing to comment on.
Author Response
Dear Reviewer,
Thank you for the notes and please find our answer in below:
- Authors should dedicate more space to the role of iron in heart failure.
We have enriched text of the manuscript with the additional information about ferroptosis role in HF. (see text in red color)
- Are there any treatments based on a ferroptosis-blocking mechanism that could be translated to acute coronary syndrome?
A few RCT were conducted, but in practice currently, there is no such treatment prescribed.

Reviewer 2 Report
This is an interesting review paper about ferroptosis as the new therapeutic target of heart diseases. This manuscript reviews the potential contribution of ferroptosis to cardiomyopathy, the mechanisms of ferroptosis, and a list of ferroptosis inhibitors. There are a few key points that the authors should address to make this manuscript suitable for publication.
Major Comments
1) The authors reviewed multiple mechanisms of ferroptosis, which could vary between cell types. Which mechanisms are specific to heart cells? This should be mentioned in the manuscript for clarify.
2) The authors also reviewed a list of ferroptosis inhibitors. Authors should further mention which agents were tested in heart cells (or not). In case the inhibitors were tested in both heart and non-cardiac cells, it will be useful to compare the efficacy of these inhibitors between cell types (in case this information is available in the literature).
3) It is important to list which heart models (in vivo, in vitro) were studied for (1) and (2). It is also helpful to summaries this information in a Table.
Minor Comments
Please move Figures S1 and S2 to the major figures. Table 1 can be shown together with the main text (instead of supplementary information).
/
Author Response
Dear Reviewer,
Thank you for the notes and please find our answer in below:
Major Comments
1) The authors reviewed multiple mechanisms of ferroptosis, which could vary between cell types. Which mechanisms are specific to heart cells? This should be mentioned in the manuscript for clarify.
Well noted, manuscript updated according to your notes (see text in red colour)
2) The authors also reviewed a list of ferroptosis inhibitors. Authors should further mention which agents were tested in heart cells (or not). In case the inhibitors were tested in both heart and non-cardiac cells, it will be useful to compare the efficacy of these inhibitors between cell types (in case this information is available in the literature).
Well noted, text of manuscript and table updated with the section Testing model and information about type of test: ex vivo, in vivo, RCT.
3) It is important to list which heart models (in vivo, in vitro) were studied for (1) and (2). It is also helpful to summaries this information in a Table.
Well notes, table updated to your recommendations.
Minor Comments
Please move Figures S1 and S2 to the major figures. Table 1 can be shown together with the main text (instead of supplementary information).
Corrected.
